# Insights into cork weathering regarding colour, chemical and cellular changes in view of outdoor applications

**Isabel Miranda, Ana Lourenço👤, Rita Simões, João Athayde, Helena Pereira👤***

Forest Research Center and Associate Laboratory TERRA, School of Agriculture, University of Lisbon, Lisboa, Portugal

* hpereira@isa.ulisboa.pt

**Data Availability Statement:** All relevant data are within the manuscript.

## Abstract

A comprehensive analysis of outdoor weathering and soil burial of cork during 1-year experiments was carried out with measurements of CIELAB color parameters, cellular observations by scanning electron microscopy, and surface chemical features analysed by ATR-FTIR and wet chemical analysis. Cork applied in outdoor conditions above and below ground retained its physical structure and integrity without signs of deterioration or fracturing. The cellular structure was maintained with some small changes at the one-cell layer at the surface, featuring cellular expansion and minute cell wall fractures. Surface color and chemistry showed distinct results for outdoor exposure and soil burial. The weathered cork surfaces acquired a lighter color while the soil buried cork surfaces became darker. With outdoor weathering, the cork polar solubles increased (13.0% vs. 7.6% o.d. mass) while a substantial decrease of lignin occurred (about 28% of the original lignin was removed) leading to a suberin-enriched cork surface. The chemical impact on lignin is therefore responsible for the surface change towards lighter colors. Soil-burial induced hydrolysis of ester bonds of suberin and xylan, and the lignin-enriched cork surface displayed a dark brown color. FTIR and wet chemical results were consistent. Overall cork showed a considerable structural and physical stability that allows its application in outdoor conditions, namely for building façades or other surfacing applications. Architects and designers should take into account the color dynamics of the cork surfaces.

## Introduction

Cork is a non-wood forest product obtained from the outer bark of cork oak trees with a sustainable exploitation process based on periodic removals along the tree's lifetime. Cork oaks (*Quercus suber* L.) have high ecological and economic importance by participating in environmental protection and biodiversity, and providing the raw material for a strong cork industrial chain [1]. Cork has an interesting combination of properties, including its chemical and biological inertness and durability that results from the material's cellular and chemical features [2]. Cork is world known as the closure for wine bottles, but many other cork products are

**Funding:** This research was funded by the Portuguese Foundation for Science and Technology (FCT) through the funding of the Forest Research Centre (UIDB/00239/2020) and the project SIZA/ETM/0050/2019. Rita Simões acknowledges a doctoral scholarship from FCT with the SUSFOR Doctoral Programme (PD/BD/128259/2016), and Ana Lourenço a research contract (DL 57/2016/CP1382/CT0007) also funded by FCT. The funder had no role in study design, data collection and analysis, decision to publish, or preparation of the manuscript.

**Competing interests:** The authors have declared that no competing interests exist

manufactured and applied as surfacing or insulation materials, namely for the construction and building sector, that also take advantage of the cork's renewable character and favorable ecological footprint [3].

Cork has been extensively studied in relation to structure, chemical composition, properties and applications, and the cork industry developed a strong technological innovative character, as compiled by Pereira [1]. However, there are no studies focusing on cork weathering, namely on the impact of light, and especially UV, humidity and rain as well as of polluted environments, *e.g.* urban spaces, and soil contact on cork materials. These aspects are important since cork products are now increasingly applied in outdoor environments as surfacing and insulation coverings. The use of cork in building façades has increased in the last years, triggered by the public attention given to some buildings with outdoor cladding with cork, *e.g.* in the Hannover and Shanghai World Exhibitions (2000 and 2010, respectively), or in outdoor installations *e.g.* in the Serpentine Gallery in London (2012). The use of cork in construction is in line with the present trend of green and sustainable building with use of materials that have a low ecological footprint [3]. This is the case of cork that has also interesting properties regarding insulation, low density and water permeability, as well as chemical inertia and durability. Knowledge on its weathering performance is therefore important for informed application in construction and architectural approaches.

Weathering is a phenomenon that occurs slowly over time, causing changes in both aspect and properties of the materials, primarily due to irreversible changes in their chemical properties. The effect of weathering and aging has been extensively studied for wood and wood-based materials in association with protection methods [4–6]. The wood cell wall polymeric structures undergo several degradation mechanisms during aging such as photolysis, thermolysis, oxidation, and hydrolysis [7, 8]. The chemical action results in the breaking down of lignin, hemicelluloses and cellulose polymers that may lead to cell wall defibrillation and fractures of the cellular structure [9–11]. The most visually striking effect is the change of colour of the wood surface which strongly impacts on the aesthetical perception, but weathering also affects the wettability of the wood surface and the surface layer strength [12, 13].

As regards cork, it is empirically known that outdoor exposure causes a change on the surface color, mostly increasing lightness. The chemical composition of cork is different from that of other plant tissues, namely of wood, since it is characterized by the presence of suberin as the main cell wall structural component (on average 42%) which is absent in wood, while including also lignin (22%), polysaccharides (20%) and extractives (16%) of both lipophilic and hydrophylic nature [14]. Suberin is an aliphatic glyceridic polyester composed of glycerol and long chain (mainly $C_{18}$) poly-functional fatty acids, ω-hydroxyacids and α,ω-diacids, with minor amounts of 1-alcohols. While it can be anticipated how lignin and polysaccharides will behave during weathering, given the many studies already carried out for wood, nothing is known on the outdoor stability of suberin.

The present research addresses this knowledge gap. Cork weathering is important for applications in outdoor environments, namely as surfacing material in building façades, with soil contact also relevant in some cases. While color changes have the highest visual impact, changes at cellular and chemical levels may also impact on the long-time cork performance. This study presents a first-time analysis of outdoor weathering and underground burial of cork during 1-year exposure regarding impact on color, chemical composition and structure of natural cork. Measurements of CIELAB color parameters were made, cellular features were observed with scanning electron microscopy, and surface chemical features were analyzed by ATR-FTIR, analytical pyrolysis and by wet chemical analysis.

## Material and methods

### Samples

Rectangular thin strips of cork were used with the approximate dimensions of 10 cm width, 30 cm length and 6.5 mm thickness, representing respectively tangential, axial, and radial directions. The cork strips were cut tangentially from good quality thin cork planks with approximately 3 cm calliper that were previously boiled following the usual industrial post-harvest processing of raw cork planks. The cork sheets were cut in the mid-section of the planks, discarding the fractions near the belly or the back of the planks. The largest surface of the cork strips corresponds to the tangential section of cork (tangential x axial).

### Outdoor field test

An outdoor field test site in the School of Agriculture campus, in Lisbon, Portugal (Lat: 38˚42'27.5"N; Lng: 9˚10'56.3"W) was used for the cork testing during one year, from March 2022 to February 2023. Lisbon has a temperate climate (Köppen Csa) characterized by hot and dry summers and cool and rainy winters. The mean value of the daily temperature in the field weathering test was 18.5˚C with the lowest temperature below 10˚C observed in January and the average maximum temperature in August (the hottest month) reaching 38.3˚C, the total rain fall was 681.7 mm (100 days with precipitation, 265 days without precipitation), and with more than 2800 yearly sunshine hours. Average UV values ranged from 3 to 6 (moderate) between October and April, and from 9 to 10 (very high) between May and September (Weatherspark.com) [15]. The soil at the site is a Vertisol from basaltic and calcareous origin, with a pH of 7.0 (in water) [16].

The outdoor testing of the samples included an outdoor weathering and a soil burial degradation test using the same cork samples, following the testing procedures for biodegradation in natural weathering and open-air burial conditions set in ASTM G7/G7M-13 [17] and ASTM G160–12 standards [18], respectively. The samples were buried in the soil in vertical position to a depth of approximately 12 cm, with the largest surface facing south, and allowing normal incidence of solar radiation over the entire exposed surface of the samples. The soil line was marked on the samples. A total of six cork samples were used.

After 12 months, the cork samples were removed from the soil, washed carefully with water to remove dust and impurities, and dried in an oven at 60˚C. The samples were subdivided in two by cutting at about 1cm above and below the soil mark, corresponding to the natural weathered samples and the buried samples, respectively. The cork surface that was analysed was the south exposed surface and no measurements were made on the opposite surface. Unexposed cork samples were kept as reference.

### Scanning electron microscopy (SEM)

Small samples were cut from the test cork samples and their south exposed tangential surface was observed by scanning electron microscopy (SEM) with a Hitachi TM 3030 Plus Tabletop at 5 kV acceleration, 30 Pa vacuum, with a Mix observation mode, without sample surface metallization, at magnifications of 600 and 1500. The images were recorded in digital format.

### Colour measurement

The surface colour was measured using the CIELAB colour system, as defined by the International Commission on Ilumination (CIE), with a chroma meter Minolta spectrophotometer CM-A145 (Osaka, Japan), following the ISO 7724 standard test method. The software generated the $L^*$, $a^*$ and $b^*$ values. $L^*$ represents the lightness or brightness (0 is black and 100

white), and $a^*$ and $b^*$ denote redness (green to red) and yellowness (blue to yellow), respectively. The six samples were measured before starting the field testing and then after each treatment (outdoor exposure and soil burial). On each sample four different positions were randomly selected for mesurement, and the average value was calculated.

The color change ($\Delta E^*$) that occurred with the outdoor exposure and soil burial in relation to the unexposed initial cork was calculated according to the following equation:

$$\Delta E^* = \sqrt{\left(\Delta L^*\right)^2 + \left(\Delta a^*\right)^2 + \left(\Delta b^*\right)^2}$$

where $\Delta L^*$, $\Delta a^*$, and $\Delta b^*$ represent the changes in $L^*$, $a^*$, and $b^*$ between the initial and final values for each sample, respectively. A low $\Delta E^*$ value means a small colour change and higher photo stability.

## ATR-FTIR spectroscopy

The surface chemical groups were analysed by attenuated total reflectance-Fourier transform infrared spectroscopy (ATR-FTIR) with spectra taken directly on the sample surface. Spectra were collected using a Perkin Elmer spectrometer 400 equipped with an ATR (attenuated total reflectance) sampling accessory attached to a diamond crystal. The spectra were acquired by accumulating 8 scans at a spectral resolution of 4 $cm^{-1}$ in absorbance mode from 1800–800 $cm^{-1}$, and standardised using the baseline method. Spectral data were analysed using the Spectragryph 1.2.15 software.

## Chemical composition

The exposed cork surfaces were manually removed by a careful shaving of the superficial layers using a sharp hand knife that took a layer with an approximate thickness of 0.5 mm from the south exposed cork surface. Given the mass needed for the wet chemical analysis, the shavings taken from the six samples were mixed into a composite sample. The cork shavings were milled in a coffee grinder, dried in an oven at 60°C, and kept for analysis. A reference sample was also prepared from unexposed cork samples.

Chemical summative analyses included determination of soluble extractives, suberin, Klason and acid soluble lignin, and the monomeric composition of polysaccharides, including quantification of neutral sugars and uronic acids. All determinations were made in duplicate.

The soluble extractible compounds were determined in a Soxhlet apparatus with a solvent sequence of increasing polarity with dichloromethane (DCM), ethanol (EtOH) and water ($H_2O$) during 6 h, 16 h and 16 h, respectively.

Suberin content was determined in the extractive-free material by use of methanolysis with sodium methoxide in absolute methanol under reflux for 3 h [19]. The reaction products were filtered, washed with methanol, the pH of extracts adjusted to 6 with sulphuric acid in methanol, and concentrated. The concentrate was suspended in water and the alcoholysis products were recovered by extraction with dichloromethane in three successive extractions. The suberin content of cork was determined as the solid mass loss after methanolysis and expressed in percent of the extractive-free cork, thereby including all the monomers solubilized from suberin, i.e. including the glycerol that remains in the aqueous phase and the long chain aliphatic components that are soluble in the dichloromethane phase. The dichloromethane extracts from the suberin depolymerisation were kept for determination of the long chain monomeric composition of suberin while the water phase was kept for determination of glycerol.

Klason and acid-soluble lignin were determined in the extracted and desuberinised materials by acid hydrolysis with 72% sulphuric acid. Klason lignin was determined as the mass of the acid-insoluble lignin and the acid-soluble lignin was determined on the filtrate by measuring absorbance at 206 nm using a UV/VIS spectrophotometer, and expressed in percent of the extractive-free cork.

The polysaccharides were analysed by quantifying the total monosaccharides released by the acid hydrolysis used for lignin determination, and expressed in percentage of the total monomers. The composition of monosaccharides and uronic acids was determined using a Dionex ICS-3000 system in HPIC-PAD and an Aminotrap plus Carbopac PA10 column (250 x 4 mm). In the conditions used, mannose was eluted with xylose, and was not out singled.

## Suberin monomeric composition

The monomeric composition of suberin was determined in aliquots from the dichloromethane extracts obtained from the suberin depolymerization. The samples were evaporated under $N_2$, derivatized by trimethysilylation of the hydroxyl and carboxyl groups into trimethylsilyl (TMS) ethers and esters, respectively, and immediately analysed by GC-MS (Agilent 5973 MSD) with the following conditions: Zebron 7HG-G015-02 column (Phenomenex, Torrance, CA, USA) (30 m, 0.25 mm; ID, 0.1 μm film thickness), flow 1 mL/min, injector 380˚C, oven temperature program: 50˚C (held 1 min), rate 10˚C/min up to 150˚C, rate of 5˚C/min up to 200˚C, 4˚C/min up to 300˚C, and rate of 10˚C/ min up to 380˚C (held 5 min). The MS source was kept at 220˚C and the electron impact mass spectra (EIMS) taken at 70 eV of energy. The area of peaks in the total ion chromatograms was integrated and their relative proportions expressed as percentage for semi-quantitative analysis. Compounds were identified as TMS derivatives by comparing their mass spectra with a GC–MS spectral library (Wiley, NIST), personal library and by comparing their fragmentation profiles with published data.

The glycerol released by the suberin depolymerisation was quantified in the aqueous layer obtained from the liquid–liquid separation of the solubilised compounds by suberin depolymerization using high-performance liquid chromatography (HPLC). The determination was made with a Dionex ICS-3000 system equipped with an electrochemical detector (Sunnyvale, CA, USA), with Aminotrap plus CarboPac SA10 anionexchange columns (Thermo Scientific, Waltham, MA, USA) and a mobile phase of an aqueous 2-nM sodium hydroxide (NaOH) solution at a flow rate of 1.0 mL/min at 25˚C.

## Composition of the lipophilic extract

Aliquots of the dichloromethane extracts obtained by the Soxhlet extraction of cork were evaporated under $N_2$ flow and dried under vacuum at room temperature overnight. Derivatization of samples and GC-MS analysis were made with the same procedure and conditions as described above for the analysis of suberin extracts.

## Composition of polar extractives

The polar extracts that were obtained by ethanol and water extraction were used to determine the content in total phenolics. The determination was carried out following the colorimetric method using Folin-Ciocalteu´s reagent [20] with a slight modification. Aliquots of the ethanol and water extracts (100 μL) were mixed with 4 mL of diluted Folin reagent (1:10 with water), kept for 30 min at room temperature, after which 4mL of 7% of $Na_2CO_3$ solution were added, vortexed and incubated in a water-bath at 45˚C for 5 min. The absorbance of the resulting blue coloured mixtures was recorded with a spectrophotometer (UV-160A Recording Spectrophotometer, SHIMADZU) at 765 nm against a blank containing only water. The total phenolic

content was determined through a standard curve of gallic acid (0.014 to 0.762 g/L) and expressed as mg of gallic acid equivalents (GAE) per g of extract and g GAE per kg of dried cork. The determination was made in three replicates.

## Analytical pyrolysis

A mass of about 110 µg of dried extractive-free and desuberized cork was weighted in a quartz boat and pyrolyzed at 550°C (1 min) in a 5150 CDS apparatus linked to a gas chromatograph from Agilent (GC 7890B) with a mass detector (5977B) at 70 eV electron impact voltage. The volatiles formed were separated in a fused capillary column ZB-1701 (60 m x 0.25 mm i.d. x 0.25 µm film thickness), using helium as the carrier gas (total flow of 1mL/min). The temperatures applied were 280°C in the interface and 270°C in the injector. The oven temperature program started at 40°C (for 4 min), raised to 70°C at a rate of 10°C min$^{-1}$, then raised to 100°C (at 5°C min$^{-1}$), to 265°C (at 3°C min$^{-1}$, held for 3 min), and to 270°C (at 5°C min$^{-1}$, held for 9 min). The cork derived compounds were identified by comparing with Wiley and NIST2014 computer libraries, and relevant literature [21]. The percentage of each compound was calculated based on the total area of the chromatogram. The percentage of p-hydroxyphenyl (H), guaiacyl (G), and syringyl (S) lignin-derived products were summed separately, and the S/G ratio and H:G:S relation were calculated.

## Statistical analysis

One-way repeated measures analysis of variance (ANOVA) was performed to compare the colour parameters of the cork samples between the unexposed sample (reference) and the samples after the 1-year outdoor exposure and soil burial. Pairwise differences were evaluated with Tukey and Duncan´s test with the statistical significance set at $p < 0.05$. Sigmaplot® (Version 11.0, Systat Software, Inc., Chicago, IL, USA) was used for all statistical analyses.

# Results

The cork samples after 1-year exposure maintained their physical integrity, without visible damages, microbial growth or material fractures, as shown in S1 Fig.

## Surface morphology

The surface of the reference cork and of samples subjected to weathering and soil burial were observed by SEM in order to have an insight into the arrangement of the one-cell layer coverage of the cork surface (Fig 1).

The cellular structure observed in the reference cork sample corresponds to the known structure of cork, as seen in the tangential section, where the cells appear as polygons, mostly hexagons, with a honeycomb arrangement. After 1-year outdoor weathering, the cellular organization was maintained, as well as the overall aspect of the cells, although cell expansion was observed to a small extent, and some cell wall degradation occurred, namely of the cell prism bases. After 1-year soil burial, the cork cellular structure also maintained its honeycomb arrangement, but also showing to a small extent cell expansion and cell wall degradation leading to little material fragments that could be observed in the cells. The cell walls showed minute fractures that were more evident in the prism bases.

## Surface color change

Cork has a light brown color that underwent clear changes after weathering in outdoor conditions and in soil burial that were conspicuous to the visual observation and distinct between

Reference　　　　Outdoor weathering　　　　Soil burial

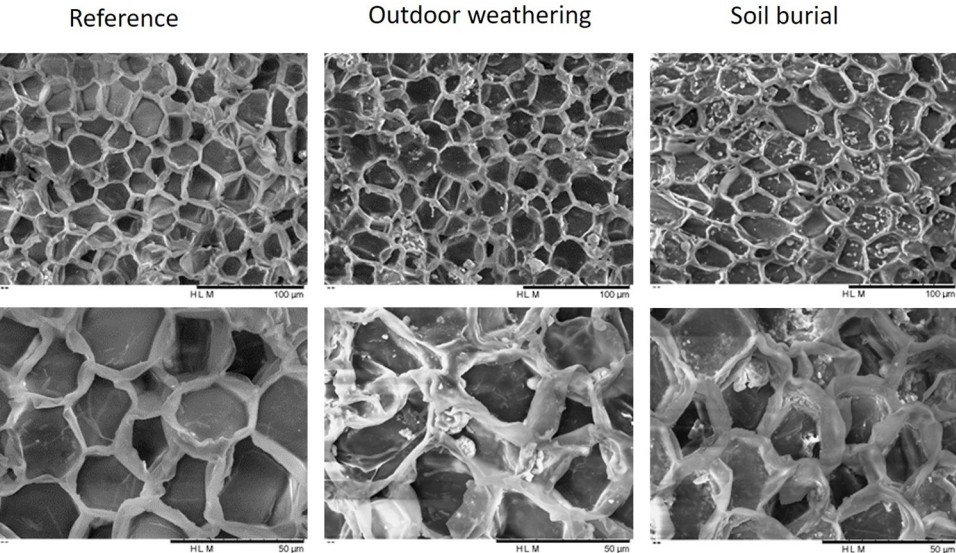

**Fig 1. SEM micrographs of the cork surface (tangential section) of the reference cork and of cork after outdoor weathering and soil burial for 12 months, at 600X magnification (top row) and at 1500X magnification (bottom row).**

the two exposure types. Outdoor weathering induced a lighter whitish color in the cork surface, while the soil burial darkened the cork. Photographic images of examples are included in Fig 2, and Table 1 shows the average CIELAB parameters of lightness $L^*$, $a^*$ and $b^*$ for the reference, outdoor exposure and soil burial samples.

In outdoor conditions, the cork surface became whiter with an increased lightness ($L^*$), while redness (reduction in $a^*$ from 13 to 6) and yellowness decreased (reduction of $b^*$ from 24 to 15), indicating that weathering tended to greening and bluing. With soil burial, $L^*$ decreased from 55.7 to 40.9, i.e. the cork surface darkened, while for $a^*$ and $b^*$ parameters the values also decreased but less when compared to the outdoor exposure, reaching 10 and 19, respectively. The statistical analysis showed that the outdoor exposed and buried samples were

Reference　　　　Outdoor weathering　　　　Soil burial

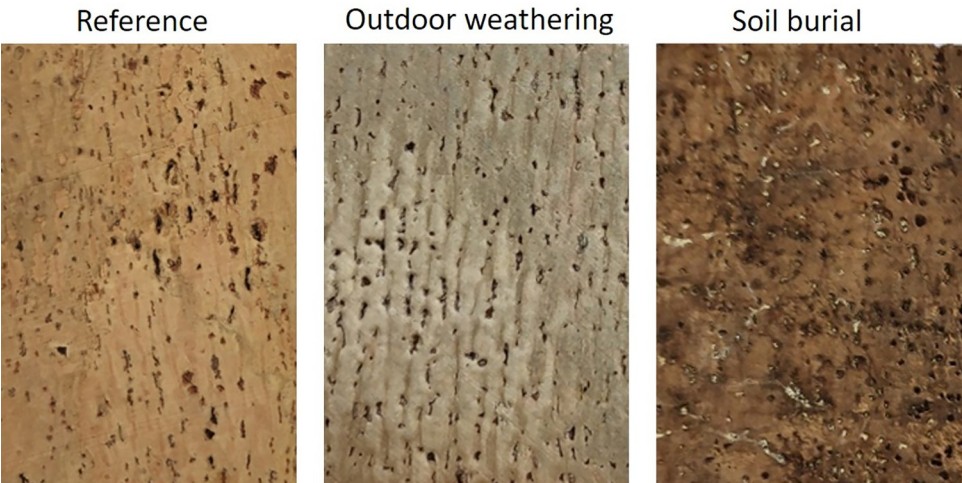

**Fig 2. Photographs of the cork surface of the reference cork and of cork samples after 1-year outdoor weathering and soil burial.**

**Table 1. CIELAB color parameters of the cork surface: In the reference cork and cork samples after 1-year outdoor weathering and soil burial.** Mean of six samples and standard deviation.

|        | Reference      | Outdoor exposure | Soil burial    |
|--------|----------------|------------------|----------------|
| L*     | 55.7 ± 2.0a    | 58.3 ± 6.7b      | 40.9 ± 3.7a    |
| a*     | 12.9 ± 1.3a    | 5.6 ± 2.5b       | 9.9 ± 1.4b     |
| b*     | 23.5 ± 1.3a    | 15.2 ± 3.6b      | 19.1 ± 2.2b    |
| ΔE     |                | 13.5 ± 3.5a      | 15.9 ± 5.4a    |

Results in a row followed by different letters are statistically significantly different (p<0.05)

significantly different from the unexposed sample for a and b, while for L* the outdoor exposed samples was different from the reference and the soil buried sample (Table 1).

## ATR-FTIR surface chemical composition

The qualitative assessment of the chemical changes occurring in the cork surface due to outdoor weathering and soil-burial was made using attenuated total reflectance Fourier transform infrared (ATR-FTIR) spectroscopy. Fig 3 compares the ATR-FTIR spectrum of the original cork surface (A) and the spectra of cork surfaces after outdoor exposure (B) and soil burial (C).

The spectrum of the original cork surface (Fig 3 Reference) has the typical features of a cork material, characterized by the presence of the suberin absorption peaks that act as a cork fingerprint. Strong absorptions at 2917 cm$^{-1}$ and 2852 cm$^{-1}$ are assigned to asymmetric and symmetric C-H stretching of the methylene group of cork suberin, respectively, and the strong band at 1739 cm$^{-1}$ is due to the stretching vibration of C = O in fatty acids and esters of suberin [16, 22–24]. These very conspicuous bands are accompanied by less defined bands at around 1635 cm$^{-1}$ for the C = C absorption in suberin long chain monomers, and at 1356 cm$^{-1}$ that reflects C-H symmetric deformations of the long aliphatic hydrocarbon chain (C–H), as well as by bands at 1230 cm$^{-1}$, 1154 cm$^{-1}$ and 718 cm$^{-1}$, corresponding to symmetric and asymmetric C-O stretching, and C-H bending associated with vinyl groups, respectively [22, 23, 25].

The suberin absorption bands may however be strongly overlapped with signals of lignin and polysaccharides when the same chemical bonds are present in the macromolecules [23, 26]. Two characteristic lignin bands were observed at 1505 cm$^{-1}$, assigned to the C = C aromatic skeletal stretching vibrations of the benzene ring, and at 1466 cm$^{-1}$, assigned to the breathing

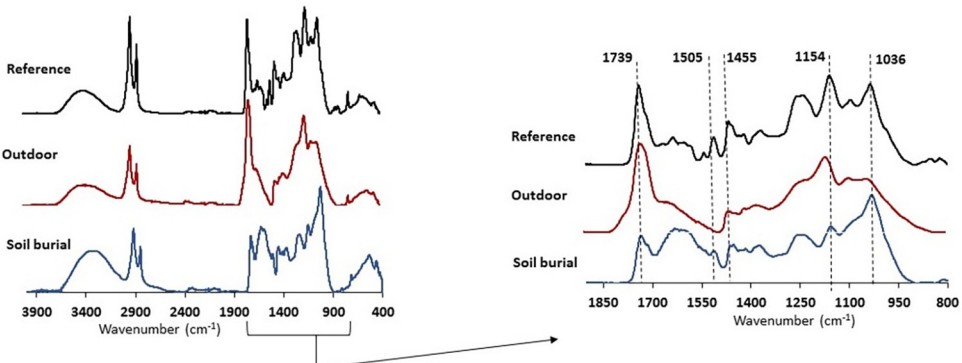

**Fig 3. FTIR-ATR spectra of the original cork surface and cork surfaces after outdoor weathering and soil burial with a spectral enlargement in the fingerprint region (1850 to 800 cm$^{-1}$).**

of S-units, as well as a band at around 1607 cm$^{-1}$ for C = C vibration in lignin. Polysaccharides contributed to absorption at 1090 cm$^{-1}$ and 1036 cm$^{-1}$ for C–H, C–O and C–OH deformations.

The ATR-FTIR spectrum of the cork surface after outdoor weathering (Fig 3 Outdoor) shows differences in relation to the natural cork spectrum that can be attributed to the chemical changes occurring at the surface. A striking difference is found in lignin-derived bands with the disappearance of the peak at 1505 cm$^{-1}$ and the decreased intensity of the peak at 1455 cm$^{-1}$, which means that degradation of lignin occurred related to the benzene ring structure. The same happened to polysaccharides with a decreased intensity and less resolution of the bands at 1036 cm$^{-1}$ and 1090 cm$^{-1}$. The characteristic suberin-related bands were present in the spectrum (at 2917 cm$^{-1}$, 2852 cm$^{-1}$ and 1739 cm$^{-1}$) although with a small difference in the relative intensity of methylene and carbonyl bands, while the other suberin-associated absorbances appeared less resolved and mostly as bands or shoulders, *e.g.* the 1635 cm$^{-1}$ peak associated to C = C in suberin monomers and the 1230 cm$^{-1}$ peak associated to C-O symmetric stretching.

After soil burial, several differences were also observed in the ATR-FTIR spectra of cork (Fig 3 Soil burial). The suberin bands at 1739 cm$^{-1}$ (C = O vibrations) and at 1154 cm$^{-1}$ (C-O vibrations) decreased their intensity, and were less resolved. The bands responsible for C-O vibrations in polysaccharides at 1090 cm$^{-1}$ and 1036 cm$^{-1}$ were less resolved in relation to the reference cork spectrum, with the peak at 1090 cm$^{-1}$ appearing as a shoulder. The buried cork samples also showed remarkable differences in the intensity of the bands attributed to lignin at 1505 cm$^{-1}$ (C = C aromatic ring stretching in lignin) which was less resolved and smaller, and at 1356 cm$^{-1}$ (C-O stretch in lignin).

## Changes in cork chemical composition

Table 2 presents the chemical composition of the reference cork and of the external layers of the cork samples after outdoor exposure and soil burial. The reference cork included 12.5% of extractives, and suberin and lignin represented, respectively, 57.4% and 22.8% of the extractive-free cork. Both outdoor exposure and soil burial induced changes in the chemical composition of the cork surface but of a distinct nature.

With outdoor weathering, the chemical changes regarded mostly the polar solubles (ethanol and water extracts) that increased substantially to 13.0% of the cork mass (7.6 in the reference cork), namely the water solubles (3.5 *vs.* 10.0%). The structural compounds showed a substantial decrease of lignin to 16.8% of the extractive-cork, leading to a significant suberin-enriched cork surface of 61.6% of the extractive-free cork. The polysaccharide composition showed a decrease in the proportion of arabinose (15.1% of monosaccharides *vs.* 24.0% in the reference cork).

After soil burial, the cork surface also showed chemical changes although of a distinct nature (Table 2). Suberin content was lower (46.9% of the extractive-free cork) and lignin content increased to 29.7%. The polysaccharide composition was similar to that of the reference cork with a small decrease in the proportion of xylose (25.5 to 22.7%) and arabinose (24.0 to 22.4%).

## Composition of extractives

Table 3 presents the chemical composition of the dichloromethane extracts solubilized from the cork surface layer of the original cork and of the cork samples after outdoor exposure and soil burial.

**Table 2. Chemical composition of the surface layer of the original cork and of the cork samples after outdoor exposure and soil burial, regarding extractives soluble in dichloromethane (DCM), ethanol (EtOH) and water (H$_2$O, in % of the cork samples), and of suberin, lignin and polysaccharides (in % of the extractive-free cork samples) as well as the monomeric composition of the polysaccharides (in % of total monosaccharides).**

| | Reference | Outdoor exposure | Soil burial |
|---|---|---|---|
| % of cork | | | |
| **Total extractives** | **12.5** | **18.6** | **11.7** |
| DCM extractives | 4.9 | 5.7 | 4.6 |
| EtOH extractives | 4.1 | 2.9 | 2.1 |
| H$_2$O extractives | 3.5 | 10.0 | 5.0 |
| % of extractive-free cork | | | |
| **Suberin** | **57.4** | **61.6** | **46.9** |
| **Total lignin** | **22.8** | **16.9** | **29.7** |
| Klason lignin | 20.1 | 13.8 | 25.9 |
| Soluble lignin | 2.7 | 3.1 | 3.8 |
| **Polysaccharides*** | **19.8** | **21.6** | **23.4** |
| Polysaccharide composition, % of total monomers | | | |
| Glucose | 41.3 | 49.8 | 43.0 |
| Xylose & mannose | 25.5 | 25.1 | 22.7 |
| Arabinose | 24.0 | 15.1 | 22.4 |
| Galactose | 7.7 | 9.0 | 9.8 |
| Rhamnose | 1.5 | 0.93 | 2.1 |

*calculated by difference

The lipophilic extractives of the reference cork included mainly terpenes (24.2% of the total), mainly the pentacyclic triterpene friedelan-3-one (15.9%). Saturated, unsaturated and substituted alkanoic acids were also present in considerable proportions (18.7% of alkanoic acids and 32.0% ω-hydroxy acids), with 3-hydroxypropanoic acid (32.9%) and hexadecanoic (6.4%), as the most represented compounds. Glycerol amounted to 5.1% and alkanols to 3.5%. Aromatics were present (4.6% of all compounds), including mainly benzoic acid and vanillin. Sterols only represented 1.3%, mainly β-sitosterol.

In weathered cork, the lipophilic extract composition was similar in what regards the families and compounds present, although with proportional differences e.g. hexadecanoic acid and friedelan-3-one increased. A similar result was obtained for cork after soil burial, with proportional differences regarding mainly an increase of triterpenes.

Table 4 presents the content of total phenolics in the ethanol and water extracts obtained from the surface layer of the cork samples under evaluation. Cork polar extracts contained a low proportion of phenolic compounds of 24.3 and 32.0% of the extract (as GAE) respectively in ethanol and water extracts, which corresponds to 1.0 and 1.1 g GAE/100 g cork. However, the content of phenolics in the surface layers of the cork samples after outdoor exposure decreased substantially to 0.2 and 0.7 g GAE/100 g of cork respectively in ethanol and water extracts. The same happened for cork samples after soil burial for which the content in soluble phenolics decreased to 0.1 and 0.2 g GAE/100 g of cork respectively in ethanol and water extracts.

## Composition of suberin

The compounds solubilised by suberin depolymerisation include the lipid monomers soluble in dichloromethane and the water-soluble glycerol that was quantified in the aqueous fraction

**Table 3. Chemical composition of the lipophilic extract obtained from dichloromethane solubilization of the surface layer of the original cork (reference) and of the cork samples after outdoor weathering and soil burial, as percentage of the total peak areas of in the GC-MS chromatogram.** Only compounds over 1% at least in one of the samples are included.

| | Reference | Outdoor exposure | Soil burial |
|---|---|---|---|
| **Alkanols** | **3.5** | **2.4** | **0.11** |
| Docosanol | 1.5 | 0.6 | - |
| Others | 2.0 | 1.7 | 0.11 |
| **Alkanoic acids** | **18.7** | **35.2** | **29.2** |
| Hexanoic acid | 3.1 | - | 1.9 |
| Tetradecanoic acid | 1.5 | 3.7 | 2.3 |
| Pentadecanoic acid | 1.3 | 1.6 | 0.9 |
| Hexadecanoic acid | 6.4 | 15.7 | 11.7 |
| Heptadecanoic acid | - | 1.2 | - |
| Octadecanoic acid | - | 2.7 | 1.8 |
| 9-Hexadecenoic acid | 1.1 | - | 2.2 |
| 9-Octadecenoic acid | 1.1 | - | 4.9 |
| Others | 4.1 | 10.4 | 3.5 |
| **a,ω-Diacids** | **1.3** | **5.6** | **0.9** |
| Butanedioic acid | 0.9 | - | 0.9 |
| Heptanedioic acid | - | 1.2 | - |
| Octanedioic acid | - | 1.2 | - |
| Nonanedioic acid | - | 2.9 | - |
| *Others* | *0.43* | *0.32* | *0.0* |
| **ω-Hydroxy acids** | **32.9** | **0.1** | **14.4** |
| 3-Hydroxypropanoic acid | 32.9 | - | 14.4 |
| **Glycerol and derivatives** | **5.1** | **4.1** | **0.4** |
| Glycerol | 5.1 | 3.8 | 0.4 |
| **Terpenes** | **24.2** | **39.8** | **45.9** |
| Friedelane-1-ene-3-one | 4.9 | 6.1 | 8.2 |
| Friedelan-3-one | 15.9 | 30.8 | 31.4 |
| Betulinic acid | 2.4 | 2.3 | 5.6 |
| Others | 1.0 | 0.7 | 0.7 |
| **Steroids** | **1.3** | **0.00** | **0.9** |
| β-sitosterol | 1.3 | - | 0.9 |
| **Aromatic compounds** | **4.6** | **3.0** | **2.1** |
| Benzoic acid | 2.5 | 1.1 | 1.8 |
| Vanillin | 1.4 | 0.8 | - |
| Others | 0.7 | 1.1 | 0.4 |
| **Others** | **0.4** | **3.7** | **0.6** |
| **Total identified** | **91.9** | **93.9** | **99.7** |

**Table 4. Phenolic content (in mg GAE/g extract) of the ethanol and water extracts obtained from the surface layer of the original cork and of the cork samples after outdoor exposure and soil burial.** Mean of three replications and standard deviation.

| | Reference | Outdoor exposure | Soil burial |
|---|---|---|---|
| **Total phenolics** (mg GAE g$^{-1}$ extract) | | | |
| EtOH extract | 242.7 ± 13.3 | 77.6 ± 19.7 | 58.6 ± 8.3 |
| H$_2$O extract | 320.1 ± 66.4 | 68.1 ± 4.2 | 118.5 ± 13.4 |

**Table 5. Suberin composition (in mass percent of extractive-free cork) including total suberin (as determined by mass loss by methanolysis), glycerol, long chain lipid compounds and aromatics solubilized by methanolysis of the surface layer of the original cork (reference) and of the cork samples after outdoor exposure and soil burial.**

|  | Reference | Outdoor exposure | Soil burial |
|---|---|---|---|
| **Total suberin** | **57.4** | **61.6** | **46.9** |
| **Glycerol** | **1.6** | **2.1** | **4.0** |
| **Long-chain lipids** | **50.4** | **52.5** | **36.9** |
| Alkanols | 0.6 | 2.6 | 0.6 |
| Alkanoic acids | 0.7 | 2.0 | 0.6 |
| a,ω-Diacids | 10.7 | 17.5 | 8.3 |
| ω-Hydroxy acids | 38.4 | 30.4 | 27.4 |
| **Aromatic compounds** | **2.3** | **1.4** | **2.9** |

of the methanolysis reaction solution. The suberin content by chemical families of its monomers in mass percent of the extractive-free cork is given in Table 5, and the monomeric composition of suberin in proportion of the total compounds determined in the dichloromethane extract obtained from the methanolysis reaction is given in Table 6.

In the reference cork, glycerol represents 1.6% of the extractive-free cork, and the major suberin long chain monomers are ω-hydroxyacids and α,ω-diacids that together accounted for 49.1% of the extractive-free cork, corresponding to 85.6% of the total suberin (Table 5). The main components (Table 6) of ω-hydroxyacids (that account for 69.0% of total monomers) are 18-hydroxyoctadec-9-enoic acid (22.3%), 22-hydroxydocosanoic acid (21.9%), and 9,10-epoxy-18-hydroxy octadecanoic acid (15.9%), and the main component of α,ω-diacids (accounting to 19.1% of total monomers) is 9-epoxi-octadecanedioic acid (10.1%).

In weathered cork, the suberin at the surface showed some compositional differences. The ω-hydroxyacids decreased and the α,ω-diacids increased (30.4% and 17.5% of the extractive-free cork respectively, Table 5). The monomeric composition (Table 6) shows a decrease of ω-hydroxyacids (51.0% of the total monomers vs. 69.0%) and a higher proportion of α,ω-diacids (29.4% vs. 19.1%). The differences in the proportion of individual monomers were the following: decrease of 9,10-epoxy-18-hydroxyoctadecanoic acid (3.1% *vs.* 15.9%), 9-epoxi-octadecanedioic acid (2.6% *vs.* 10.1%) and of 18-hydroxyoctadec-9-enoic acid (21.7% *vs.* 1.4%), while the proportion of shorter chain length monomers increased for 9-hydroxynonanoic acid (14.0% *vs.* 0.4%), and octanedioic acid (9.1% *vs.* 0.3%). The cork subjected to soil burial and compared to the reference cork (Table 5) contained more glycerol (4.0% of the extractive-free cork), similar amount of α,ω-diacids and less ω-hydroxyacids (8.3% and 27.4% respectively). The monomeric composition of the suberin (Table 6) was similar to that of the reference cork.

## Composition of lignin

The pyrograms obtained from the extractive-free and desuberinised cork samples are represented in Fig 4 and the results from the pyrolysis analysis are summarised in Table 7. The pyrograms were similar with clearly defined peaks corresponding to the thermally released volatile compounds during pyrolysis. The lignin content was slightly lower for the outdoor exposed sample and higher for the buried sample but lignin monomeric composition differed with less guaiacyl units in the outdoor weathered sample and less syringyl units in the soil buried sample, leading respectively to higher and lower S/G (0.11 in the reference cork and, respectively, 0.14 and 0.03 in the outdoor exposed and soild buried samples).

**Table 6. Suberin composition of the surface layer of the original cork (reference) and of the cork samples after outdoor exposure and soil burial (as percentage of the total peak areas of chromatogram of the methnolysis dichloromethane extract).** Only compounds above 1% in at least one sample are included.

| | Reference | Outdoor exposure | Soil burial |
|---|---|---|---|
| **Alkanols** | **1.0** | **4.3** | **1.3** |
| 2-(2-methoxyethyl)-1-heptanol | - | 1.7 | - |
| 3-(2-methoxyethyl)-1-octanol | - | 1.8 | - |
| Others | 1.0 | 0.8 | 1.3 |
| **Alkanoic acids** | **1.3** | **3.3** | **1.4** |
| **a,ω-Diacids** | **19.1** | **29.4** | **19.4** |
| Hexanodioic acid | - | 1.3 | - |
| Heptanedioic acid | - | 2.6 | - |
| Octanedioic acid | - | 9.1 | - |
| Nonanedioic acid | - | 3.4 | - |
| 9-Octadecenedioic acid | 2.8 | 0.16 | 2.0 |
| Docosadenedioic acid | 1.5 | 1.7 | 1.5 |
| 9-Epoxi-octadecanedioic acid | 10.1 | 2.6 | 10.4 |
| 9,10-Dihydroxyoctadecanedioic acid (threo) | 1.3 | 0.61 | 1.5 |
| 9,10-Dihydroxyoctadecanedioic acid | 1.7 | 4.3 | 2.4 |
| Others | 1.7 | 3.6 | 1.6 |
| **ω-Hydroxy acids** | **69.0** | **51.0** | **64.0** |
| 9-Hydroxynonanoic acid | 0.40 | 14.0 | - |
| 22-Hydroxydocosanoic acid | 21.9 | 22.9 | 21.5 |
| 24-Hydroxytetracosanoic acid | 2.2 | 1.6 | 1.9 |
| 18-Hydroxyoctadec-9-enoic acid | 21.7 | 1.4 | 15.1 |
| 9,10-Epoxy-18-hydroxyoctadecanoic acid | 15.9 | 3.1 | 17.8 |
| 18-Hydroxyoctadecanoic acid, 9(10) hydroxy-10(9)-methoxy | 1.8 | 0.7 | 1.9 |
| 9,10,18-Trihydroxyoctadecanoic acid | 2.5 | 4.2 | 3.6 |
| *Others* | *2.6* | *3.1* | *2.2* |
| **Aromatic compounds** | **4.0** | **2.4** | **6.8** |
| Methyl ferulate | 2.5 | 1.1 | 3.7 |
| **Total identified** | **94.4** | **90.4** | **92.9** |

## Discussion

### Structure and color

It is interesting that the cork material retained its physical integrity and cellular characteristics after 1-year outdoor weathering and soil burial (S1 Fig). The surface of the samples maintained the typical cork cellular structure observed in the tangential section, with a honeycomb arrangement of mostly hexagonal cells [27]. The small cell expansion and the minute cell wall degradation that were observed (Fig 1) should derive from a combined effect of swelling due to temperature and water, as it is known to occur in cork, although the reported values were obtained in harsher conditions [28, 29], as well as to some loss of stiffness of the cell walls oriented perpendicular to the tangential surface, caused by chemical changes, as it will be discussed below. However, the properties of the cork material should be retained e.g. physical integrity and insulation properties, allowing the outdoor performance of cork.

The most striking effect of weathering and soil burial was the change of surface color (Fig 2) as expected from what is known for wood, and the empirical observations of existing outdoor applied cork materials. Outdoor weathering induced in the cork surface a lighter whitish color with changes governed more by the white/grey ($L^*$) component, compared to the red/

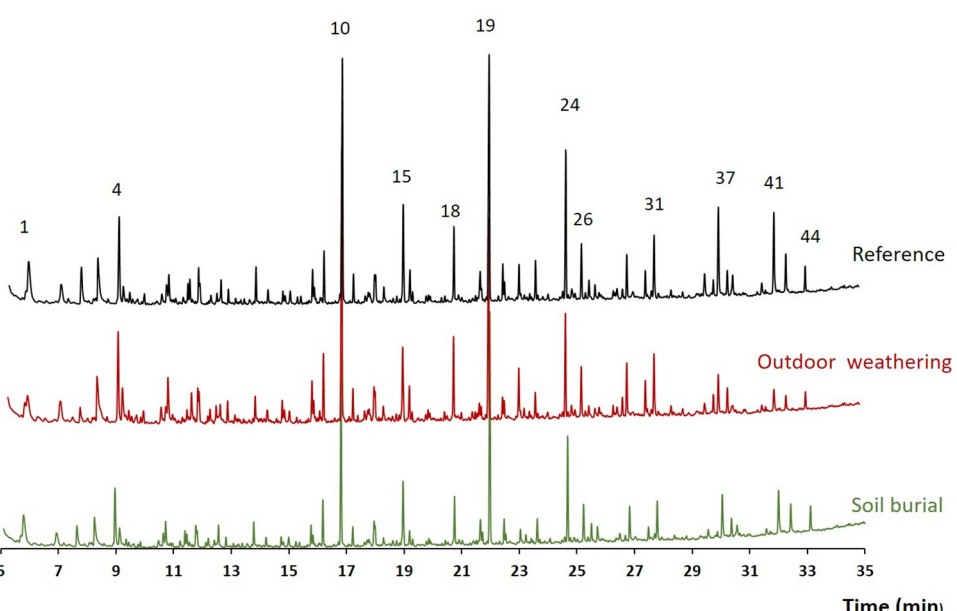

**Fig 4. Pyrograms of extractive-free and desuberinised cork samples from the surface layer of the original cork (reference) and of the cork samples after outdoor exposure and soil burial.** 1) 2-oxo-propanal; 4) 1-hydroxy-2-propanone; 10) guaiacol; 15) 4-methylguaiacol; 18) *p*-4-ethylguaiacol; 19) 4-vinylguaiacol; 24) *trans* isoeugenol; 26) vanillin; 31) guaiacylacetone; 37) dihydroconiferyl alcohol; 41) *trans* coniferyl alcohol; 44) ferulic acid methyl ester. The chemical structures of these compounds are given in S2 Fig.

green (*a\**) and yellow/blue (*b\**) color components (Table 1). This may be explained by the photodegradation of the cork cell wall components, namely of lignin [30]. In fact, visible and UV irradiation oxidizes lignin with the reduction of paraquinone to hydroquinone, causing photo bleaching effects, as it is well documented for wood surfaces [31, 32].

The soil burial darkened the cork but with smaller changes in relation to the reference cork (Table 1). Underground, cell wall decay processes are driven mainly by the action of water and anoxic biological agents that preferentially attack polysaccharides via enzymatic degradation, mostly involving hemicelluloses, while lignin is considered to be more resistant to biological decay, largely due to its highly stable structure of cross-linked polyphenol subunits [33–35]. The darkening of oak wood in a wet environment was reported as due from reaction between iron compounds present in water and the wood tannins [36]. The dark coloration of pine wood (*Pinus sylvestris* L.) samples found in a wet archaeological site might result from chemical reduction of sulphates to hydrogen sulphide, and production of dark-coloured sulphides by bacteria of the genus *Clostridium* [37].

**Table 7. Lignin composition (in % of the total pyrogram peak area) and ratios given by the analytical pyrolysis of extractive-free and desuberinised samples from the surface layer of the original cork and of the cork samples after outdoor exposure and soil burial.**

|  | Reference | Outdoor exposure | Soil burial |
|---|---|---|---|
| **Total lignin** | 49.9 | 47.8 | 57.0 |
| ***p*-Hydroxyphenyl units (H)** | 4.2 | 5.4 | 5.2 |
| **Guaiacyl units (G)** | 41.2 | 37.0 | 50.1 |
| **Syringyl units (S)** | 4.5 | 5.4 | 1.7 |
| **S/G ratio** | 0.11 | 0.14 | 0.03 |
| **H:G:S relation** | 1: 9.8: 1.1 | 1: 6.8: 1.0 | 1: 9.7: 0.3 |

It is interesting to notice that the colour changes were limited to the external cork layer and the underlying cork retained its original colour. Therefore, the removal of the external layer (*e. g.* by sanding) may be applied in case the original colour is wanted, in a process similar to what is often done for wood surfaces. However, the integration of cork color dynamics during weathering into the architectural and design creative process is probably more advantageous regarding sustainability, and will allow an enrichment in the chromatic palette of the built environment.

## Chemical impact on cork components

The chemical changes occurring in the cork surface due to outdoor weathering and soil-burial were assessed by ATR-FTIR spectroscopy (Fig 3) and by wet chemical analysis of the cork surface layer (Table 2) with compositional analysis of its extractives (Tables 3 and 4), suberin (Tables 5 and 6) and lignin (Table 7).

The spectrum of the original cork surface has the typical features of cork, with the fingerprint of suberin absorption, highlighted by the conspicuous bands at 2917 $cm^{-1}$ and 2852 $cm^{-1}$ (C-H stretching), and 1739 $cm^{-1}$ (C = O in fatty acids and esters) [1]. Characteristic lignin bands are present at 1505 $cm^{-1}$ (C = C aromatic) and 1466 $cm^{-1}$ (S-units), as well as a band at around 1607 $cm^{-1}$ (C = C vibration), while bands associated to polysaccharides occur at 1090 $cm^{-1}$ and 1036 $cm^{-1}$ (C–H, C–O and C–OH deformations).

After outdoor weathering, the ATR-FTIR spectrum of the cork surface shows that lignin degradation occurred by the disappearance of the 1505 $cm^{-1}$ band and the decrease of the 1455 $cm^{-1}$ band. Some impact was seen also on polysaccharides mainly on the bands at 1036 $cm^{-1}$ and 1090 $cm^{-1}$, as well as on suberin with a small difference in the relative intensity of methylene and carbonyl bands, and less resolution in other bands, *e.g.* 1635 $cm^{-1}$ band associated to (mid chain C = C) and 1230 $cm^{-1}$ band (C-O symmetric stretching).

Overall the ATR-FTIR spectra indicate that UV light induces a decrease of lignin content and its chemical modification (Fig 3). The substantial increase of the C = O band at 1737 $cm^{-1}$ may result from the formation of lignin chromophores with carbonyl groups in the photo-oxidized lignin [11, 38–40]. The intensity increase at 1730 and 1650 $cm^{-1}$ may also derive from C = O groups of unconjugated ketones in hemicelluloses and p-OH substituted arylketones and quinines in lignin, which indicate oxidation of hemicelluloses and lignin during weathering. Weathering was reported to promote destruction of the lignin-hemicellulose matrix, resulting in the leaching of hemicelluloses from the cell walls [41, 42], which agrees with the findings of the decrease of carbohydrate-related bands. Regarding suberin, the presence of the characteristic main bands shows the weathering stability of suberin, although the lower intensity and less resolution of the other bands suggest that chemical changes occurred in the long chain monomers, namely at the mid-chain functionalization *e.g.* C = C and epoxide.

After soil burial, the ATR-FTIR spectra of cork showed an impact on suberin by cleavage of intermonomeric ester bonds, as indicated by the decrease in intensity of the 1739 $cm^{-1}$ (C = O vibrations) and at 1154 $cm^{-1}$ (C-O vibrations) peaks. The decreased intensity at 1739 $cm^{-1}$, as well as the band at 1616 $cm^{-1}$ are also indicative of a modification in hemicelluloses since the xylan backbone contains C = O bonds in acetyl and carbonyl groups. The observed decrease is well known and due to reduction reactions of carbonyl functional groups [9, 35, 43, 44]. A structural alteration of hemicelluloses was also suggested by the bands responsible for C-O vibrations in polysaccharides at 1090 $cm^{-1}$ and 1036 $cm^{-1}$, which were less resolved, and the peak at 1090 $cm^{-1}$ appearing as a shoulder. The buried cork samples also showed remarkable differences in the intensity of the lignin bands at 1505 $cm^{-1}$ (C = C aromatic ring stretching) which was less resolved and smaller, and 1356 $cm^{-1}$ (C-O stretch). These results suggest that

the chemical changes of cork after soil-burial are due to hydrolysis of ester bonds of suberin and xylan, with a simultaneous alteration in lignin bands.

The chemical composition of the reference cork given by wet chemical procedures met the chemical parameters known for cork [1, 14], namely regarding the summative analysis (Table 2), as well as the compositional features of extractives, suberin and lignin (Tables 2–7). Both outdoor exposure and soil burial induced changes in the chemical composition of the cork surface but of a distinct nature.

With outdoor weathering, the polar solubles increased substantially (by an absolute amount of 5.4% of the cork mass, corresponding to an increase of 71% in relation to the original cork) that originated from the degradation of lignin. In fact, lignin decreased significantly (in an absolute amount of 6.0% of the extractive-free cork mass, corresponding to a 26% removal of the original lignin), therefore leading to a suberin-enriched cork surface with a suberin:lignin ratio of 3.66 (the reference cork had a suberin:lignin ratio of 2.52). Hydrolysis of hemicelluloses also occurred, especially of arabinose, while cellulose remained stable as given by the glucose content, leading to an increased glucose-to-non-glucose ratio (1.0 *vs*. 0.7 in the original cork). The results obtained by wet chemical analysis are clearly in line with the ATR-FTIR spectral analysis of the cork surface (Fig 3).

It is well known from studies on wood weathering, that the major effect is delignification, with lignin and hemicelluloses being easily decomposed by UV irradiation, while cellulose is assumed to be protected from photodegradation because the UV light is preferentially absorbed by lignin [6, 45]. In fact, lignin absorbs about 80–95% of the UV light due to the presence of chromophores and aromatic rings, making it more prone to decomposition by photo-oxidation reactions [46, 47]. The interesting and novel result is that suberin remains stable with weathering, thereby being a chemical asset for the well-known long-term stability of cork. Therefore, the cork surface change towards lighter colors (Fig 1) is due to the chemical impact of weathering on lignin.

After soil burial, the chemical changes of the cork surface were different from those of outdoor weathering (Table 2), and an impact was found on suberin. Suberin was lost by an absolute amount of 10.5% of the extractive-free cork mass (corresponding to a removal of about 18% of the suberin in the original cork) while lignin remained unchanged and therefore its proportion in the cork surface increased, leading to a marked decrease of the suberin:lignin ratio to 1.58. The polysaccharide fraction was also affected with a small decrease of hemicelluloses by hydrolysis of xylose and arabinose units, although to a limited extent (the glucose-to-non-glucose ratio was 0.75). Studies on wood degradation induced by burial also refer that hemicelluloses are the most affected, while cellulose content remains almost stable over time [48–50]. Again the wet chemical results accord with the spectral observation (Fig 3), as discussed above, of a clear decrease of suberin-related bands as well as those of hemicelluloses, and show that surface ATR-FTIR spectroscopy is a very useful tool to evaluate chemical changes. In the case of cork after soil-burial, these changes refer to hydrolysis of ester bonds of suberin and xylan, which lead to a dark brown color of the lignin-enriched surface (Fig 1).

## Impact on chemical composition of cork components

The chemical impact of outdoor weathering and soil burial may occur on the composition of the cork components, adding to the changes in content that were described in the previous section. This possibility was addressed by the analysis of the composition of lipophilic and polar extractives (Tables 3 and 4) and the monomeric composition of suberin (Tables 5 and 6) and of lignin (Table 7).

The composition of the lipophilic extractives of the reference cork, including mainly terpenes and alkanoic acids, and also glycerol, aromatics and sterols, is overall in line with previous values given for cork extractives although highly variable proportions were reported [51, 52]. In weathered cork, the lipophilic extract composition was similar in what regards the families and compounds present, and the small proportional differences in some compounds *e.g.* hexadecanoic acid and friedelan-3-one are in line with the natural variability of cork. The unchanged composition of the lipophilic extractives was to be expected given the fact that suberin content was not altered by weathering. In the cork after soil burial, the composition was in general similar to the reference cork but with proportional differences regarding mainly an increase of triterpenes. Given that the content in dichloromethane extracts remained stable (Table 2), this result may be explained by a combined effect of the cork surface washing and soil compound reaction of the new lipophilic solubles generated from suberin depolymerization.

The content of total phenolics in the ethanol and water extracts of cork were in accordance with reported values [53], but decreased substantially after outdoor exposure and after soil burial. These results point out to a leaching of polar extractives from the cork surface with the contact of environmental water.

As regards suberin composition of the reference cork sample, the major monomers are ω-hydroxyacids and α,ω-diacids, in line of what is reported for cork [1].

In weathered cork, the suberin at the surface showed some compositional differences with a proportional decrease of ω-hydroxyacids and an increase of α,ω-diacids (the ratio ω-hydroxyacids: α,ω-diacids was 1.74 *vs*. 3.6 in the reference cork), and also differences in the proportion of individual monomers that indicate a chemical impact in the mid-chain substitution. For instance, in the monomers with epoxide and C = C mid chain substitution, mid-chain cleavage may occur (9-epoxy-18-hydroxyoctadecanoic acid, 9-epoxi-octadecane-dioic acid and 18-hydroxyoctadec-9-enoic acid decreased), giving a higher proportion of C9 and C8 monomers. This midchain cleavage reaction of the long chain monomers of suberin under UV radiation does not induce degradation of the suberin polymer and therefore a decrease in the overall suberin content did not occur (Table 2). This is in agreement with the ATR-FTIR results showing peak alterations regarding C = C and epoxy bonds in suberin (Fig 3).

In the cork subjected to soil burial, the suberin composition was similar to that of the reference cork with a ratio ω-hydroxyacids: α,ω-diacids of 3.3. Taking into account that the cork contained less suberin (Table 2), this result suggests a depolymerization by cleavage of the inter-monomeric ester bonds and the removal of the monomers, while the remaining suberin retains its macromolecular composition. This is also in accordance with the ATR-FTIR results showing a decrease of ester bonds in the cork surface (Fig 3).

Regarding the lignin in cork, there is a predominance of guaiacyl with a small proportion of syringyl and *p*-hydroxyphenyl units, as previously determined [54–56]. In the weathered cork, some degradation of guaiacyl units occurred, mainly demethoxylation, that could explain the slight increase of H-units [40]. In the cork soil burial, there was a degradation of the more reactive syringyl units, since lignin degradation is faster under aerobic conditions [57], and the breakdown by fungi, namely by white rot fungi, is proposed to have as primary step a demethoxylation process [58].

## Conclusions

Cork applied in outdoor conditions above and below ground during one-year exposure retained its physical structure and integrity without signs of material deterioration or

fracturing. The cellular structure was maintained with some small changes at the one-cell layer at the surface, featuring cellular expansion and minute cell wall fractures.

Impact occurred on the cork surface as regards color and chemistry with distinct results for outdoor exposure and soil burial. The weathered cork surfaces acquired a lighter color given mainly by lignin degradation and partial solubilization, while suberin retained its polymeric characteristics with some chemical changes in the long-chain monomers. The soil buried cork surfaces acquired a darker color mainly derived from an increased lignin content due to polysaccharide degradation and suberin depolymerization.

Overall cork showed a considerable structural and physical stability that allows its application in outdoor conditions, namely for building façades or other surfacing applications. Certainly the present one-year experimental results should be complemented with longer duration testing. Architects and designers should take into account the color dynamics of the cork surfaces along time.

## Supporting information

**S1 Fig.** Photograph of one unexposed cork sample and of one 1-year exposed cork sample with outdoor exposure (right) and soil burial (left).
(JPG)

**S2 Fig. Chemical structures of the main peaks represented in the pryrograms of the cork samples (Fig 4).**
(PDF)

## Acknowledgments

We thank Joaquina Silva for her technical assistance in chemical analysis. The cork sheets were supplied by the cork industry Amorim & Irmãos, S.A. (Santa Maria de Lamas, Portugal). We thank Jorge Gominho and Antonio V. Marques for providing their personal mass spectra library for GC-MS analysis.

## Author Contributions

**Conceptualization:** Isabel Miranda, Helena Pereira.

**Data curation:** Isabel Miranda.

**Formal analysis:** Ana Lourenço, Rita Simões.

**Investigation:** Ana Lourenço, Rita Simões, João Athayde.

**Methodology:** Isabel Miranda, Ana Lourenço.

**Resources:** Isabel Miranda, Helena Pereira.

**Supervision:** Helena Pereira.

**Validation:** Isabel Miranda, Helena Pereira.

**Visualization:** Helena Pereira.

**Writing – original draft:** Isabel Miranda, Helena Pereira.

**Writing – review & editing:** Isabel Miranda, Ana Lourenço, Helena Pereira.

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
