## [Decision Letter · Decision Letter 0]

26 Jan 2024

PONE-D-23-38130Insights into cork weathering regarding colour, chemical and cellular changes in view of outdoor applicationsPLOS ONE

Dear Dr. Pereira,

Thank you for submitting your manuscript to PLOS ONE. After careful consideration, we feel that it has merit but does not fully meet PLOS ONE’s publication criteria as it currently stands. Therefore, we invite you to submit a revised version of the manuscript that addresses the points raised during the review process.Please go thoroughly through reviewers' comments and make needed corrections. However, except the minor revisions, the second reviewer asked for some major changes, as follows:

Major comments

1. Line 131: I suggest that the cork samples collected on day 0 would serve as a more appropriate control for comparisons of color and chemical composition after the 1-year experiment. Please provide additional details regarding the use of the side of the cork plank with less exposure to light as the reference.

I would like you to consider this suggestion as I find it very suitable.

2. I believe this study included the replications during their experiments, but I didn’t see the statistical analysis.

I would like you to consider the idea of showing those results as well.

3. Line 171: The cork samples were dried in an oven at 60 °C before microscopic and chemical analysis. Could this temperature be considered a harsh treatment for cork, potentially causing changes in color and chemical composition?

Please explain in details.

Additionally, I would like you to consider some changes in your discussion and conclusion - put emphasis on applicability (architecture, environment protection, green housing) and change (or add) keywords accordingly.

We look forward to receiving your revised manuscript.

Kind regards,

Iskra Alexandra Nola

Academic Editor

PLOS ONE

“This research was funded by the Portuguese Foundation for Science and Technology (FCT) through the funding of the Forest Research Centre (UIDB/00239/2020) and the project SIZA/ETM/0050/2019. Rita Simões acknowledges a doctoral scholarship from FCT with the SUSFOR Doctoral Programme (PD/BD/128259/2016), and Ana Lourenço a research contract (DL 57/2016/CP1382/CT0007) also funded by FCT.”

5. We note that your Data Availability Statement is currently as follows: [All relevant data are within the manuscript and its Supporting Information files.]

Reviewers' comments:

Reviewer's Responses to Questions

**Comments to the Author**

1. Is the manuscript technically sound, and do the data support the conclusions?

Reviewer #1: Yes

Reviewer #2: Partly

Reviewer #3: Yes

2. Has the statistical analysis been performed appropriately and rigorously? 

Reviewer #1: Yes

Reviewer #2: No

Reviewer #3: Yes

3. Have the authors made all data underlying the findings in their manuscript fully available?

Reviewer #1: Yes

Reviewer #2: Yes

Reviewer #3: Yes

4. Is the manuscript presented in an intelligible fashion and written in standard English?

Reviewer #1: No

Reviewer #2: No

Reviewer #3: Yes

5. Review Comments to the Author

Reviewer #1: Add the importance of study and relevance this study to international readers.

Why this study should be done?

what are the limitation of this study?

double check the citation and the references list. Make sure all the citations were included in the references.

The references should be adjusted as the journal guideline.

Check the grammatical errors.

In conclusion you should add the future research prospect related the present study.

Reviewer #2: In this manuscript, the authors investigate the effects of 1-year outdoor weathering and soil burial on the color and chemical composition of cork. This study found that except the color changes, the cork maintains its physical structure and integrity when applied in outdoor conditions, including both above and beneath the ground. This is an interesting study focusing on the not well-explored aspects of cork weathering. Their results provide evidence of the potential for using cork as a green material in building facades. However, in my opinion, this study lacks proper control and statistical analysis. Once these issues are addressed, I believe this paper will be a valuable addition to the PLOS ONE journal.

Major comments

1. Line 131: I suggest that the cork samples collected on day 0 would serve as a more appropriate control for comparisons of color and chemical composition after the 1-year experiment. Please provide additional details regarding the use of the side of the cork plank with less exposure to light as the reference.

2. I believe this study included the replications during their experiments, but I didn’t see the statistical analysis.

3. Line 171: The cork samples were dried in an oven at 60 °C before microscopic and chemical analysis. Could this temperature be considered a harsh treatment for cork, potentially causing changes in color and chemical composition?

Minor comments:

1. Line 32: The letter "ç" in the word "façades" is not used in English. I believe this non-English character appears in all instances of "façade" throughout the manuscript.

2. Lines 90 and 93: The full names of "CIELAB" and "FTIR-ART" should be provided when these terms first appear in the manuscript.

3. Line 264: I recommend providing a photo of the cork samples after one year of exposure to demonstrate their physical integrity as the first figure.

4. In figure 3 and 4, I recommend adding illustrations of the important chemical structures next to their corresponding peaks.

5. Line 585: I believe the correct word should be “from” instead of “form”.

Reviewer #3: I recommend acceptance this manuscript as it is valuable and related to one of the most world challenges which is the pollution of the environment. As the author showed the different application of the cork and how it will be useful especially in the architecture and building field. In addition, cork showed a considerable structural and physical stability that allows its application in outdoor conditions.

6. PLOS authors have the option to publish the peer review history of their article (what does this mean?). If published, this will include your full peer review and any attached files.

Reviewer #1: No

Reviewer #2: No

Reviewer #3: No

---

## [Author Response · Author response to Decision Letter 0]

31 Jan 2024

The answer to all reviewers's comments are in the dociment "Response to reviewers"

---

## [Decision Letter · Decision Letter 1]

14 Mar 2024

Insights into cork weathering regarding colour, chemical and cellular changes in view of outdoor applications

PONE-D-23-38130R1

Dear Dr. Pereira,

We’re pleased to inform you that your manuscript has been judged scientifically suitable for publication and will be formally accepted for publication once it meets all outstanding technical requirements.

Kind regards,

Iskra Alexandra Nola

Academic Editor

PLOS ONE

Please read carefully 2nd reviewer's comment to the authors. If there is any issue to be solved please let us know.

Reviewers' comments:

Reviewer's Responses to Questions

**Comments to the Author**

1. If the authors have adequately addressed your comments raised in a previous round of review and you feel that this manuscript is now acceptable for publication, you may indicate that here to bypass the “Comments to the Author” section, enter your conflict of interest statement in the “Confidential to Editor” section, and submit your "Accept" recommendation.

Reviewer #2: All comments have been addressed

Reviewer #3: All comments have been addressed

2. Is the manuscript technically sound, and do the data support the conclusions?

Reviewer #2: Yes

Reviewer #3: Yes

3. Has the statistical analysis been performed appropriately and rigorously? 

Reviewer #2: Yes

Reviewer #3: Yes

4. Have the authors made all data underlying the findings in their manuscript fully available?

Reviewer #2: Yes

Reviewer #3: Yes

5. Is the manuscript presented in an intelligible fashion and written in standard English?

Reviewer #2: Yes

Reviewer #3: No

6. Review Comments to the Author

Reviewer #2: Upon careful examination of the revised manuscript and the authors' responses to the comments provided during the initial review, I believe the authors have adequately addressed all major concerns and most minor concerns. Regarding minor comment 1, my intention was not to suggest that the word “facade” is improper in English. Rather, I aimed to inquire whether the letter “ç” is commonly used in formal English writing. As the authors mentioned, this letter is indeed of French origin. I would suggest that the authors or the publication team at PLOS One verify its appropriateness in this context. Nonetheless, this is a minor issue. Once this matter is resolved, I believe the paper is ready for publication.

Reviewer #3: I recommend to accept this manuscript. All the comments have been addressed by the authors. In addition, the statistical analysis performed appropriately and rigorously.

7. PLOS authors have the option to publish the peer review history of their article (what does this mean?). If published, this will include your full peer review and any attached files.

Reviewer #2: No

Reviewer #3: No

---

## [Editor Report · Acceptance letter]

27 Mar 2024

PONE-D-23-38130R1 

PLOS ONE

Dear Dr. Pereira, 

I'm pleased to inform you that your manuscript has been deemed suitable for publication in PLOS ONE. Congratulations! Your manuscript is now being handed over to our production team.

Kind regards, 

on behalf of

Dr. Iskra Alexandra Nola 

Academic Editor

PLOS ONE